# A qualitative, grounded theory exploration of the determinants of self-care behavior among Indian patients with a lived experience of chronic heart failure

Deepak Y. Kamath[1,2], K. B. Bhuvana[1]*, Luke Joshua Salazar[3], Kiron Varghese[4], Anant Kamath[5], Jyoti Idiculla[6], Prem Pais[2], Shruthi Kulkarni[6], Bradi B. Granger[7], Denis Xavier[1,2]

1 Department of Pharmacology, St. John's Medical College, Bengaluru, Karnataka, India, 2 Division of Clinical Research and Training, St. John's Research Institute, Koramangala, Bengaluru, India, 3 Department of Psychiatry, St. John's Medical College Bengaluru, Karnataka, India, 4 Department of Cardiology, St. John's Medical College, Bengaluru, Karnataka, India, 5 School of Development, Azim Premji University, Bengaluru, Karnataka, India, 6 Department of Internal Medicine, St. John's Medical College, Bengaluru, Karnataka, India, 7 Duke University School of Nursing, Durham, NC, United States of America

* bhuvana.bvn@gmail.com

**Data Availability Statement:** The codebook with all excerpts and codes are available on Figshare at https://figshare.com/s/001fb961c5056fd36a34.

## Abstract

### Background

Prior reports have documented extremely poor adherence to evidence-based medications among South Asian patients with established chronic cardiovascular diseases. Treatment adherence is now considered a part of the 'self-care' process, the determinants of which have not been adequately explored or explained among South Asian patients with chronic heart failure (CHF). Our objective was to qualitatively ascertain the determinants of the self-care process among Indian patients with a lived experience of heart failure.

### Methods

We conducted in-depth interviews (audio-recorded) among 22 purposively sampled patients living with chronic heart failure, diagnosed at least 4 weeks prior to the interview and 17 caregivers (n = 39) in a tertiary care teaching hospital in Southern India. We employed an inductive analytical approach using Charmaz's constructivist grounded theory. Initial line-by-line coding and categorization was followed by memo writing, reflexive analysis after interviewing and analyzing four, eight and twelve patients, and at each stage further theoretical sampling was carried out until we reached thematic saturation. We used NVivo ver. 12 to analyze and organize data.

### Results

The mean age of our patients was 61 years and they represented 5 Indian states and spoke seven languages, distributed across socio-economic strata and literacy levels. We classified self-care determinants into 3 broad, simple categories and defined underlying themes

The interview guide is available at https://doi.org/
10.6084/m9.figshare.11176187.v1.

**Funding:** This study was funded by the India
Alliance - Department of Biotechnology (DBT)/
Wellcome Trust as an early career fellowship to Dr.
Deepak Y. Kamath (KDY), ref.no. IA/CPHE/15/1/
502053. The funders had no role in study design,
data collection and analysis, decision to publish, or
preparation of the manuscript.

**Competing interests:** The authors have declared
that no competing interests exist.

namely, negative determinants (passivity, entrenched beliefs, negative affect, lack of knowledge, financial difficulties, and fatalism), intermediate factors (patient expectations, provider/hospital hopping) and facilitators or positive self-care determinants (intrinsic and extrinsic facilitators). Gender and the cultural background of patients' upbringing appear to shape these determinants, thereby affecting self-care decision making in chronic heart failure.

## Conclusion

We have empirically described a unique set of self-care determinants among Indian chronic heart failure patients, which in turn are shaped by economic and socio-cultural factors. Assessing for and addressing these determinants during clinical interactions through multifactorial approaches may help improve self-care among Indian CHF patients, thus improving treatment adherence and clinical outcomes.

## Introduction

"*There is, still, an Indian way of thinking.*"–A. K. Ramanujan, MacArthur fellow, Poet, and scholar of Indian literature.

   In South Asia, chronic heart failure (CHF) in recent years has transitioned from a syndrome caused predominantly by rheumatic heart disease to that caused by conventional cardiovascular risk factors that are well-known in developed countries [1]. South Asian patients tend to be younger on average, are more likely to present in NYHA class IV and like their Western counterparts, have a high co-morbidity and pill burden [2]. Non-adherence to treatments increases the risk of re-hospitalizations and mortality [3]. The Prospective Urban Rural Epidemiology (PURE) community cohort study demonstrated that 80.2% were not on any secondary prevention medications following an established stroke or coronary artery disease in the low-income country cohort. Adherence to statins was 3.3%, low-dose aspirin was 8.8%, beta-blockers 9.7% and ACE Inhibitors or Angiotensin Receptor Blockers (ARBs) 5.2% respectively among patients in low-income countries after a median of 2 years following the index event [4]. The problem of sub-optimal medication adherence in heart failure (HF) and other cardiovascular diseases has not been explored from the perspective of the broader concept of self-care among Indian patients. The Theory of Self-care of Chronic Illness, a type of middle-range theory of self-care, for cardiovascular disease, proposed by Riegel and Stromberg [5], defines self-care as having 3 components. These are, monitoring (recognizing and correctly interpreting symptoms and signs, "body listening"), maintenance (adherence to therapy and follow-ups) and management (responding to changing symptoms/signs) respectively. Interventions in Western countries combining approaches such as task shifting and remote monitoring to improve self-management among patients with heart failure reduced the composite risk of HF hospitalization or all-cause death (HR, 0.80 [95% CI, 0.71–0.89]) and the risk of HF hospitalization alone (HR, 0.80 [95% CI, 0.69–0.92]), and improved 12-month HF-related quality of life (standardized mean difference, 0.15 [95% CI, 0.00–0.30]) [6].

   A proposed model for decision making in chronic disease self-care is the naturalistic decision-making framework, which explains decision making in the context of real-world situations that patients find themselves in (e.g., missing information, competing goals, time stress, coping with change, etc.) that impacts situational awareness and the comprehension of the

significance of a specific situation [7]. Cultural factors and beliefs are also determinants of the quality of self-care in heart failure [8]. There exist lacunae with respect to beliefs and values peculiar to Indian patients, and how these beliefs shape self-care decision making and behavior. What is also debated is whether there are common cultural factors affecting selfcare processes across geographies. Our patients have their own distinctive explanations and perceptions of their chronic illness experience and the practice (or the lack of) of self-care in their everyday lives. Different factors, principally, gender, the geography and society of their up-bringing, shape patients beliefs, worldview and in-turn habits and behavior [9–11]. Hence decision-making frameworks need to accommodate this 'lived reality'. The objective of this study is to qualitatively understand the principal factors affecting self-care among Indian chronic heart failure patients and propose that at the foundation are socially constructed processes rooted in cultural factors and realized in behavioral traits.

## Materials and methods

### Design

We carried out a qualitative exploration of self-care behaviour and decision-making processes among patients with CHF, taking caregiver narratives also into account. The qualitative study comprised in-depth interviews with patients and their principal caregiver.

### Setting, investigator characteristics

We undertook the study in the Cardiology and Internal Medicine wards of St. John's Medical College Hospital, a tertiary care, teaching, non-profit hospital in South India. The hospital while located in a metropolitan area, also receives patients from semi-urban and rural areas from several Indian states. Ethics committee approval was obtained from the Institutional Ethics Committee, St. John's Medical College, with reference number 124/2017. Informed consent forms were translated into vernacular languages by a professional translating agency and approved by the ethics committee. Written informed consent was obtained from all participants, including semi-literate and illiterate participants in accordance with the Indian Council of Medical Research's (ICMR) National Ethical Guidelines for Biomedical and Health Research involving Human Participants 2017.

The credentials of the investigators who carried out data analysis are detailed in a prior published paper on facilitators of selfcare [12]. The principal interviewers (KDY and BKB) and data analysts are medically trained doctors with a specialization in clinical pharmacology and a research focus in cardiovascular disease (CVD) prevention. KDY and BKB, with the help of a research nurse (for translating languages) conducted all interviews. DX is a medically trained pharmacologist with a research focus in CVD prevention, KV is professor and head of cardiology, while PP, JI and SK are professors in internal medicine, LSJ is assistant professor of psychiatry at St. John's Hospital, Bengaluru. AK is a social scientist and economist, and BB is professor of nursing faculty at Duke University. DX, SK, JI, BB, KV and PP reviewed the protocol, were involved in planning the study and reviewed the manuscript. KDY, BKB, BB, LSJ and AK reviewed the memo and codebooks after KDY had completed coding at multiple stages and provided their feedback on themes and reviewed the manuscript.

Since the interviewers and investigators are from a higher socio-economic group and have an urban up-bringing, they were initially pre-disposed to viewing the self-care problem from a bio-medical perspective, rather than from a cultural or social perspective. Therefore, after 3 interviews by KDY, the interviewers agreed to modify their interviewing viewpoint to accommodate these perspectives in consonance with the social constructivist approach [13].

## Eligibility criteria and sampling

Consenting patients over 18 years of age with a clinical diagnosis of chronic heart failure (NYHA class II–IV) for ≥ 4 weeks prior to index hospitalization for acute decompensation of symptoms were eligible for inclusion. We excluded patients who were unable to provide coherent verbal information due to any medical reason *and* those who had no caregiver who could be interviewed. The 'principal caregiver' was defined as the family member or individual most involved in helping the patient manage the illness, as identified by the patient.

The purposive sampling strategy included a cross section of patients from 5 sub-classes of socio-economic status (Kuppuswamy's scale) [14], 2 levels of health literacy assessed using a 3-item brief health literacy questionnaire that classifies health literacy as high or marginal/low [15], self-reported medication-taking for one month (regular/irregular/stopped) prior to index hospitalization and gender. In addition, we captured basic clinical and demographic data and selfcare practices on a data collection form.

## Study procedures and recruitment

We conducted in-depth, face-to-face interviews of patients and their principal caregiver.

Riegel's theory of self-care of chronic illness [5] informed the development of the interview guide. Therefore, the questions mainly pertained to–knowledge of the condition, symptom and sign monitoring, medications procurement and management, adherence to diet and fluid restrictions, coping with changing symptoms, response to such changes and the management plan in case of acute deterioration. Caregiver interviews focused on the role played by caregivers in patient self-care, coping with evolving, fluctuating situations and caring for their own health. We used probe questions to ascertain or to clarify patient's beliefs pertaining to any of the aspects of self-care. While formal piloting of the guide was not carried out, few probe questions were added after the first 2 interviews. These probe questions were mainly added to elicit responses to ascertain the reasons for certain self-care behaviour patterns. No repeat interviews were carried out. The detailed interview guide has been shared on Figshare [16].

The interviews were conducted by the bedside, in languages that the patients were comfortable in (English, Kannada, Hindi, Konkani, Tamil, Telugu and Malayalam) and recorded using an audio recorder. The patients were offered the choice of having the interview conducted in a secluded office, but all except 2 patients opted for a bedside interview. Investigators made brief field notes after the interview on salient aspects such as their emotional disposition during the interview, openness to answer questions, lucidity and the presence of family members. The interviews were transcribed verbatim and then translated into English by research assistants proficient in respective languages and finally verified by one of 2 investigators (KDY, BKB) for content accuracy. Memos were made after the interview by KDY for all information rich interviews, especially noting and linking the role of gender, geography of upbringing, health literacy and socio-economic circumstances with important emerging themes affecting self-care. Interviews lasted for an average of 19 minutes 35 seconds (maximum, 37 minutes 55 seconds and minimum, 13 mins 13 seconds). We did not have an opportunity to return transcripts to participants for their comments.

Adherence to HF medications were assessed at baseline using a single-item question, "over the last month, did you take your heart medications regularly as prescribed"? Responses were recorded as 'regular/irregular/stopped'.

## Data analysis

We followed Charmaz's approach to grounded theory analysis [13] and performed a stepwise inductive analysis of the data. As a first step, data were coded inductively, line-by-line, using a content analysis approach. As a second step, after analyzing three interviews, codes were

classified into the following eight groups: self-care facilitators and barriers, attitudes (toward their condition and treatment), beliefs, knowledge, physical and psycho-social consequences of chronic illness and management practices, to make analysis convenient as more codes accrued. Common emergent themes were identified, and the interviewers focused on these themes. After completing four and twelve patient interviews, KDY carried out a reflexive analysis, including additional literature searches to support the link between emerging self-care determinants and their socio-cultural links. We decided to saturate the sample with patients from rural areas with lower health literacy, while attempting to ensure a gender balance.

We also generated comparison diagrams using NVivo, of 'similar' and 'contrasting' cases. An example for such a diagram is as illustrated in Fig 1, where we compared the codes and memos of a female patient from a rural, low socio-economic background with low health literacy and poor adherence history with the case of a male patient from an urban, high socio-economic background with a history of optimal adherence. 4 pairs of such information rich comparisons and memo notes were made. Memo notes helped us link consistently recurring themes with socio-economic and cultural determinants. The coding tree was prepared as summarized in Table 3. Data elements were coded by the first author. The codebook and memo reviews were carried out by 2 other investigators (BKB, LSJ). Coding densities were used to identify recurring themes. The investigators (DK, BKB, LSJ) concurred that theoretical saturation for facilitators was attained after interviewing 22 patients and 17 caregivers. BB and AK provided feedback on the themes and categories at multiple stages of analysis. Data analysis was done using NVivo version 12. The investigators have not had the opportunity to discuss study findings with the participants but will do so when they visit for out-patient follow-ups in future.

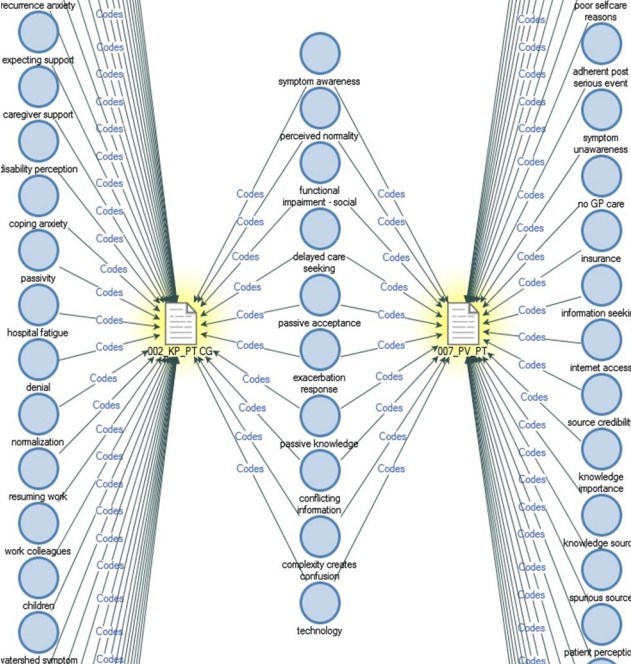

**Fig 1. Comparison diagram of codes common and divergent between 2 patients representing 'extreme cases'.** Patient 002, a young female patient with low health literacy of rural background and patient 007, an elderly gentleman from an urban location with high health literacy. Codes such as delayed care-seeking and passive acceptance of information were common to these 'extreme cases'. In case of Patient 002, delayed care seeking was due to the culturally determined behavioural trait of passivity and in case of Patient 007, due to the perception of self as an obstacle.

## Results

We approached 27 patients from March 2018 to July 2018, of whom 3 refused, on account of being too ill to speak and their caregivers being unavailable at the time of our visit. Of the 24 patients approached, 15 were patient-caregiver dyadic (n = 30) interviews. 2 patients who consented were not in a position to speak, one due to post-stroke dysarthria and the other due to fatigue, so we interviewed only their caregivers, while 7 patients did not have a caregiver or caregiver was not available during the interview (patients without caregiver = 2; caregiver was not present at the time of the interview = 5) and therefore, we could only interview the patient. Thus, 22 patients and 17 caregivers in total were interviewed (n = 39).

Patients were from 5 states of India, of which 4 were from Southern India (Karnataka, Tamil Nadu, Andhra Pradesh and Kerala) and 1 from Eastern India (West Bengal). Demographic and clinical characteristics of patients are presented in **Table 1**. We interviewed 39 participants in total. The mean age of patients was 61.2 (±13.4), with 9 (40.9%) female patients. 7 (41.2%) caregivers were male, while 10 (58.8%) were female. 17 (77.1%) patients were from the lower-middle, upper-lower, or low socio-economic strata (SES), while 5 (22.7%) were from upper-middle and high strata. Most 14 (63.6%) had low health literacy. 5 (22.7%) reported being non-adherent with medications in the past one month, with 2 (9.1%) having stopped medications completely and 3 (13.6%) reported taking medications irregularly.13 (59%) reported monitoring, but not documenting fluid intake daily. Only 4 (18.8%) were regular with physician specified out-patient follow-ups and 8 (36.3%) were aware of basic heart failure symptoms and were monitoring them.

**Table 1. Demographic characteristics, etiology of heart failure and self-care practices of the patient sample.**

| Variables | N = 22, n (%) |
|---|---|
| **Age** (Mean, SD) | 61 (13.4) |
| **Gender;** Female | 9 (40.9) |
| **Socio-economic Status (SES)** | |
| High | 3 (13.6) |
| Upper-Middle | 2 (9.1) |
| Lower-Middle | 9 (40.9) |
| Upper-Lower | 6 (27.2) |
| Low | 2 (9) |
| **Health Literacy;** Low | 14 (63.6) |
| **Etiology** | |
| Rheumatic | 1 (4.5) |
| Ischemic | 13 (59.1) |
| DCM | 4 (18.1) |
| Hypertensive | 4 (18.1) |
| **Medication Adherence;** Non-adherent | 5 (22.7) |
| **Self-Care Practices;** Regular practice | |
| Symptom monitoring | 8 (36.3) |
| Regular physician specified follow-up | 4 (18.8) |
| Fluid intake monitoring | 13 (59) |
| Pedal edema check | 6 (27.7) |
| Home blood pressure monitoring | 0 (0) |
| Daily weight monitoring | 5 (22.7) |

### Key emerging themes affecting selfcare

 **I) Negative determinants.**   Themes/ factors that consistently have a negative influence on any one or a combination of monitoring, maintenance and management. **Table 2** lists more excerpts for each code.

(i)   *Entrenched beliefs or notions* (*supporting codes*–symptom/disease causal abstractions, obstacle self-perception, ageing self-perception, non-physiological (unnatural) interventions, 'powerful' medications, trust deficit (in the healthcare system), denial or normalization, fatalism).

 **Symptom/disease causal abstractions**- Patients demonstrated a tendency to make associations or interpretation of the symptoms they were having and associating it with either one or more of somatic, dietary, or other environmental causes. These 'causal associations' were beliefs that had become entrenched. Similar abstract associations were made between treatments and symptom relief or exacerbation.

 [(Female, upper-middle SES, high health literacy): "this one, for me if this BP (blood pressure) clears,"; (KDY interviewer): "hmm"; (Patient): "it's like half of the disease is cleared." (Female, low SES, low health literacy, caregiver account): "When she eats what she is allergic to, she gets abdominal distension, she will have breathlessness, again we will take her to hospital."]

 **Obstacle self-perception**–Several patients desisted from reporting increasing discomfort or symptomatic severity to caregivers since they felt that they were obstacles in their family's routine lives. There were instances where early action in situations of progressively worsening breathlessness or edema could have been taken, but the patients did not want to report it to the caregivers.

 [(Male, low SES, low health literacy): "No.. If I go (die) that will be a relief, at least whoever are there, they can stay peacefully, if people like me exist, it's difficult (crying)."]

 **Ageing self-perception**–Some patients believed they had a cardiac problem due to ageing or that their symptoms were simply due to the ageing process, while not attributing it to other risk factors such as hypertension or smoking.

 [(Male, high SES, high health literacy): "Then as I grew old. . . (describes heart problem as a consequence)."]

 **Non-physiological (unnatural) interventions**–Patients expressed concerns that 'modern medicines' were 'not natural' and caused 'side-effects', while some patients viewed water restriction as non-physiological.

 [(Male, upper-lower SES, low health literacy): "Practically (normally), I'm not drinking water like that." (Male, high SES, high health literacy): "I believe more in homeopathy and naturopathy, ayurvedic than allopathy. BECAUSE THERE ARE LESS SIDE EFFECTS THERE (emphasis)."]

 **'Powerful' medications**–Some patients equated receiving injections with receiving 'powerful medications' giving symptomatic relief. They prioritised such 'instant gratification' treatment over long-term preventive care, going to the extent of demanding unrelated injections (such as ranitidine, an acid suppressing drug) from care providers for acute dyspnoea.

 [(Male, upper-middle SES, high health literacy): "I go straight to them (healthcare personnel at office medical centre) and tell them "Give me an injection". I show them my arm and ask them to inject (an acid suppressant for dyspnoea)."]

 **Trust deficit (in the healthcare system)**–Educated patients from urban areas expressed a sense of distrust with private for-profit healthcare, leading them to constantly doubt the diagnosis or the veracity of the efficacy of the interventions being given.

**Table 2. Themes that emerged as negative determinants of self-care.**

| Themes | Codes | Extracts |
|---|---|---|
| *Entrenched beliefs or notions* | Symptom/ disease causal abstractions | • "She feels that when she passes motion her breathlessness will automatically stop." (caregiver account on patient's explanation for breathlessness)<br>• "When her indigestion comes down (sic) her breathlessness also will come down (sic)." (caregiver)<br>• "My eyes will show me the first symptom (of heart disease). I will be fine, suddenly I will get blurring of vision." (patient)<br>• "Yes, like homeopathy. Like lemon and other things, I drink, they can reduce my secretions and breathlessness." (patient) |
| | Obstacle self-perception | • "They (children) are struggling to bear (sic) their own family; how will they bear me along with that?" (patient)<br>• "I don't want to trouble anybody. That is it." (patient)"She feels that if she makes us keep coming to her, she is troubling us." (caregiver) |
| | Ageing self-perception | • "Your age also has increased, you (patient) had (sic) to maintain with these medications only." (patient) |
| | Non-physiological (unnatural) interventions | • "I was depending more on homeopathy because there are less side effects." (patient)<br>• "Otherwise earlier, I used to drink a litre of water for lunch, dinner, breakfast." (patient) |
| | 'Powerful' medications | • "Sir, give me a Rantac (Ranitidine) injection please, that's all (for breathlessness)." (patient)<br>• "I go straight to them (healthcare personnel at office medical centre) and tell them–"Give me an injection". I show them my arm and ask them to inject (an acid suppressant for dyspnoea)." (patient) |
| | Trust deficit (in the healthcare system) | • "If you give (pay bills), they (for-profit hospital) will use you. If you are giving you should keep giving, but they will not give it back." (patient)<br>• "When they said heart problem is there (diagnosis conveyed by a physician at a for profit hospital), we didn't believe that." (caregiver) |
| | Denial or normalization | • "Nothing like that (major cardiac event) happened." (patient)<br>• "For me, for that (heart problem) I don't have any problem." (patient)<br>• "First when heart problem came (sic), we thought it's not there, heart problem was not there." (caregiver) |
| | Fatalism | • "My ideas have gone now ma. I am seeing (sic) for the way when God will call me." (patient) |
| *Passivity* | Maintenance passivity | • "He (patient, alcohol dependent) by himself will not take any medications (laughs), I will only give." (caregiver)<br>• "I am not aware about the expenses, I just come along with them and they take care of me. If they buy and give me something to eat, I will eat." (patient) |
| | One-way compliance | • Interviewer (probe): "So basically, you're following blindly what the doctor advices you? Patient: "Yes, yes" |
| | Passive receipt of information | • "No. Nobody told me. Even the doctor didn't tell me." (patient)<br>• "Something (heart disease related information), suddenly I find something on the TV or read in the newspaper." (patient) |
| *Lack of knowledge* | Lacking knowledge of diagnosis | • "But we did not know that I had a severe heart problem." (patient)<br>• Interviewer: "This, this now your heart is weak, you know that don't you?" Patient: "Yes, I found out after coming here."<br>• "There they will tell me that I have problems, then only I will come to know that I have problems." (patient) |
| | Lacking knowledge of warning symptoms | • "About illness, I don't know anything." (patient)<br>• "What symptoms. I don't know exactly about symptoms which can predict heart problem." (patient) |
| *Negative emotions/ affect* | Hopelessness | • "His lot of actions (duties), are pending (sic) which he is feeling helpless." (caregiver)<br>• "I'm fed up with my life." (patient) |
| | Coping anxiety | • "Till when I'm there somehow I have to manage." (anxious disposition; patient with no caregiver)<br>• "It has changed. I'm thinking will I be able to earn something tomorrow." (patient) |
| | Anxiety associated with stigma of chronic disease/ medication | • "As soon as I go near the box (pillbox), I feel bad." (patient)<br>• "If I go there to the container to take my tablet and come back, I feel somewhat mentally unsatisfied." (patient) |
| *Financial difficulties* | - | • "As days went now my financial status has gone down." (patient)<br>• "But we don't have enough money to take care." (caregiver)<br>• "No, no. I don't have any control over my food. I don't have enough money to do so. That is the fact. What to do." (patient)<br>• "More than this if I survive, I should be financially stable, but I'm not." (patient without caregivers) |

[(Male, high SES, high health literacy, caregiver account): "If you give (pay bills), they (for-profit hospital) will use you. If you are giving you should keep giving, but they will not give it back." (Male, high SES, high health literacy): "(Emphasis) We should have cross opinion and then take decisions because today things gone into more commercial (sic). Things are not as moral." (Female, low SES, low health literacy): "When they said heart problem is there (diagnosis conveyed by a physician at a for profit hospital), we didn't believe that."]

**Denial or normalization**–Many patients expectedly entered a denial phase when they were first told of a cardiac problem, affecting adherence in the first few months. In some patients, this denial persisted over time.

[(Male, upper-middle SES, high health literacy): "I did not get any heart attack." (the patient has CHF post-NSTEMI)] (Female, low SES, low health literacy): "When they said heart problem is there, we did not believe that, again we went to XX (another) Hospital."]

**Fatalism**–Some patients expressed a sense of inescapable pre-determined destiny guiding outcomes, some expressing this through divine connotations and others through the 'fruits' of past actions.

[(Male, upper-lower SES, low health literacy): "My ideas have gone now ma (sic). I am seeing (sic) for the way when God will call me."]

(ii) *Passivity* (*supporting codes*- maintenance passivity, one-way compliance, passive information receipt) 'Passivity' is a behavioral trait that emerged from data which we characterize as a lack of active participation and a preference for shifting responsibility for most or all of the self-care process to either the caregiver/ family or provider, with the patient playing little to no active role except to passively follow the physician's advice. We found passivity in the following situations:

**Maintenance passivity**- *relying on the caregiver for most or all aspects of selfcare monitoring, maintenance and activities of daily living*. We encountered several cases in NYHA II/III, where the caregiver had to closely oversee all aspects of self-care, activities of daily living or both.

[(Female, upper-middle SES, high health literacy): "(Irritation) All the three, I don't do anything, I just sit quietly. He (husband) does it (overseeing fluid and diet restrictions; everyday medication management). Not me."]

**One-way compliance**- We encountered patients who had doubts regarding several aspects of their treatment (ranging from medications to lifestyle modification) at clinic follow-ups, but had not ventured to ask their treating cardiologist/ physician, instead telling us that they would do whatever the physician asked them to do [(Female, high SES, low health literacy): "No. Nobody told me. Even the doctor didn't tell me."]

**Passive receipt of information**–Strikingly, few female patients in the group reported pro-actively seeking information about their condition. Rather, most female patients were passive recipients of health information mainly from physicians, or from media sources (richer patients).

[(Female, low SES, low health literacy): "Nothing, I have never thought about asking (about my health) to anyone. I will lie idle at home."]

(iii) *Lack of knowledge* (*supporting codes*- lacking knowledge of diagnosis, lacking knowledge of warning symptoms, lacking knowledge of medication subsidy schemes).
All patients recruited into the study had lived through the heart failure experience. Yet, patients from the lower SES and those with poor health literacy demonstrated an absolute lack of knowledge and situational awareness. Patients from the higher SES were aware of their 'heart problem' but were mostly unaware of aspects related to monitoring. The lack of knowledge also extended to awareness of government out-patient medicine subsidy

schemes, which could have ensured better long-term adherence among the lower socio-economic status patients.

[(Male, low SES, low health literacy): "I did not know that I had a heart problem, all I had was body pain and back pain."]

(iv) *Financial difficulties.* Financial difficulties were explained by patients in different ways. This included, (a) taking on debt or loans for treatment ("We have lots of debts, only debts! Elder son has also taken some loan."), (b) daily wage earners expressing concerns about supporting their family ("I need to earn something to support my family."), (c) lack of health insurance, (d) anticipating and coping with future health related financial problems ("More than this if I survive, I should be financially stable, but I'm not."), and (e) financially constrained to purchase healthy food ("No, no. I don't have any control over my food. I don't have enough money to do so. That is the fact. What to do.").

(v) *Negative emotions/affect* (*supporting codes-* hopelessness, coping anxiety, anxiety associated with stigma of chronic disease/ medication)

We captured a range of negative emotions/ affect associated with the presence of chronic illness and medication intake.

**Hopelessness**-[(Male, low SES, low health literacy): "If I'm saying I should die soon with my own mouth (sic) then imagine how much I'm facing (cries)."]

**Coping anxiety**- (all expressed in an anxious tone of voice & expression) pertaining to coping with activities of daily living and earnings for the future.

[(Female, rich SES, low health literacy): "I will be thinking, if I must wait at school (workplace) or go back home, will my family be able to manage my condition."]

**Anxiety associated with stigma of chronic disease/ medications**- Some female patients explicitly expressed being stigmatised either by the fact that they had a chronic condition or were on lifelong treatments.

[(Female, upper-middle SES, high health literacy): "I don't show my operated leg (amputation, diabetic foot ulcer) also to anyone. I don't show it to my children." "I don't take it (insulin injections) in front of my children. I don't take it in front of others." (Caregiver): "She can't see others when she takes medicines."].

**II) Intermediate factors.** These are distinct factors that are partly shaped by the demographic background of the patient, that may not directly have an obvious negative effect on self-care, but nonetheless modulate the selfcare process. Two instances of negative intermediate factors are given below.

(i) *Patient expectations* (*supporting codes-* expectation of symptom relief, recurrence despite compliance) Patient's expectations play an important role in adherence; patients from rural areas expect quick symptomatic relief and constantly weigh the trade-off between the money spent on hospital expenses and the benefits gained in terms of symptomatic relief and feeling well. Patients consistently expressed futility with treatments or a sense of helplessness when expectations were thwarted.

**Expectation of symptom relief.** [(Male, low SES, low health literacy): "I spent nearly 900 rupees on that day, and it is a waste. How much I would have struggled to get (earn) that 900 rupees. Either the breathlessness should go down (sic), or the stomach pain should go down (sic). Both were not happening. Still, it is the same. So that doctor is not good."]

**Recurrence despite compliance.** [(Female, upper-middle SES, high health literacy): "We were following the doctor's advice." DB (interviewer): "Hmm." (Patient): "INSPITE of that, the same thing has happened."]

(ii) ***Provider or hospital hopping.*** Patients from rural areas had transferred from one hospital to another in search of symptom relief on the advice of their village neighbors. If relief was not obtained, they went to another hospital, at times changing 3–4 hospitals consecutively and accruing large bills before coming to a tertiary care center. Sometimes, patients switched hospitals to avail of free or subsidized treatment schemes, with the result that a relationship was never established with any physician.

[(Female, low SES, low health literacy, caregiver account): "We were fed up taking her to that hospital, so we took her to a hospital at Krishnagiri."]

**III) Positive determinants.**   As published in an earlier report [12], we classified factors positively affecting self-care into 'intrinsic' (patient traits) and 'extrinsic' (external determinants) respectively. Intrinsic traits included situational awareness, self-efficacy, feeling and expressing gratitude, resilience, spiritual invocation and support-seeking behavior. Extrinsic traits, shaped or facilitated by the environment, included financial security and caregiver support, company of children, ease of healthcare access, trust in provider/hospital, supportive environment and recognizing the importance of knowledge.

Table 3 summarizes the negative determinants, intermediate factors and positive determinants.

## Memo on gender, geographic, societal factors and social discourses shaping self-care

Passivity was a trait observed uniformly among all female patients. We observed that female patients particularly tended to rely entirely on caregivers (mostly husbands or in their absence, sons or brothers, and daughters) for managing their medications, dietary oversight and for their activities of daily living. Among male patients, passivity was seen among males with negative affect (such as those who expressed hopelessness) or males with alcohol dependence, or low educational attainment, or a combination of these factors. Males were more reliant on women for self-care management and dietary oversight than for activities of daily living and managing medications.

Care seeking behavior impacting self-care management was shaped considerably by geographic location of residence and literacy levels. Patients from rural areas and lower socio-economic backgrounds, illiterate or semi-literate patients reported relying heavily on their neighbors and family members opinions in the same village for a 'good hospital or doctor'. The notion of a good hospital was generally shaped by a history of symptomatic relief obtained by patients upon treatment. This translated into a **behavior of hospital hopping or doctor hopping** from one small private facility or government facility to another until the patient reached the large tertiary care hospital (either public or private), with instances of catastrophic expenditures incurred among uninsured families. Urban, well-educated patients with internet access had established a trusted relationship with a single hospital and doctor. However, we documented these patients having a constant mistrust of several aspects of the modern medical system. These included aspects related to 'for-profit private hospitals', 'side-effects of modern medicines' and a lack of reliable health information due to temporally changing evidence pertaining to interventions that patients read in newspaper or social media reports. In contrast, rural patients were far more trusting of physicians and providers but expected early symptomatic relief and **had difficulty understanding that they were faced with a long-term condition for which treatments were required indefinitely**.

Abstract symptom, disease and treatment associations/ notions cutting across gender, socio-economic status, literacy and health literacy levels were documented. Patients whose families were in financial distress expressed more negative affect than insured, financially well-off individuals.

**Table 3. Coding tree- summary of important categories and their supporting themes and codes.**

| Category | Themes | Codes |
|---|---|---|
| **Negative Determinants** | Entrenched Beliefs | Symptom/ disease causal abstractions |
| | | Obstacle self-perception |
| | | Ageing self-perception |
| | | Non-physiological (unnatural) interventions |
| | | 'Powerful' medications |
| | | Trust deficit (in the healthcare system) |
| | | Denial or normalization |
| | | Fatalism |
| | Passivity | Maintenance passivity |
| | | One-way compliance |
| | | Passive receipt of information |
| | Lack of knowledge | Lacking knowledge of diagnosis |
| | | Lacking knowledge of warning symptoms |
| | | Lacking knowledge of medication subsidy schemes |
| | Negative emotions/ affect | Hopelessness |
| | | Coping anxiety |
| | | Anxiety associated with stigma of chronic disease/ medication |
| | Financial difficulties | Debt burden |
| | | Income shortfall |
| | | Lack of insurance |
| | | Future treatment expenses |
| | | Purchasing healthy food |
| **Intermediate Factors** | Patient expectations | Expecting symptom relief |
| | | Recurrence despite compliance |
| | - | Provider/ hospital hopping |
| **Positive Determinants (Facilitators)** | Intrinsic facilitators | Situational awareness |
| | | Self-efficacy |
| | | Expressing gratitude |
| | | Resilience |
| | | Spiritual Invocation |
| | | Support-seeking behaviour |
| | Extrinsic facilitators | Financial security |
| | | Caregiver support |
| | | Company of children |
| | | Ease of healthcare access |
| | | Trust in provider/hospital |
| | | Supportive environment |
| | | Recognizing the importance of knowledge |

If we were to contrast the urban rich versus the urban and rural poor, the theme that emerges is an inability to have a clear plan for long term care (**long term planning deficit**).

## Discussion

We set out to understand the determinants of self-care among patients with chronic heart failure using a qualitative, grounded theory approach. We found that self-care behaviour in Indian heart failure patients is shaped by multiple complex factors and not by financial constraints or a

lack of knowledge alone. These factors, for instance, various forms of entrenched beliefs (such as obstacle self-perception, abstract causal associations, 'powerful' medications and perception of the ailing self) and passivity, which negatively affect monitoring and maintenance and issues related to trust in the provider and the general healthcare system which affect maintenance and management (by doubting the effectiveness of treatments and delaying care seeking), are novel findings of this study with roots in culture, geographic and socio-economic circumstances. Conversely, patients who considered their health as their own responsibility and took the initiative to engage in selfcare displayed, overall, better adherence and selfcare. The presence of children in the household or neighbourhood and utilizing spiritual beliefs are key novel facilitators [12]. These novel determinants are compatible with the situation-specific theory of decision making in self-care in heart failure [17]. Notions around spirituality, fatalism and harbouring doubts about prescribed medications (which led to a lower commitment to adhere in future) played a role and have been identified in prior reports [18]. A systematic review of quantitative studies exploring factors determining self-care in heart failure identified only clinically diagnosed depression as a predictor of sub-optimal self-care in heart failure [19].

## How negative determinants affected the self-care process

Patients who were unaware of having a heart-related illness, tended to mistakenly attribute their symptoms to a disorder of the gastro-intestinal tract (most frequently), or as a corollary, believed that certain diets or certain actions (e.g., 'passing motions (stools)') relieved breathlessness. Consequently, patients believed that eliminating or controlling the purported 'cause' or the 'aggravating factor' would result in the 'cure' of the illness–thus reinforcing a sense that the underlying illness has a short-term course. Such entrenched beliefs, compounded by the knowledge deficit, had the effect of patients not engaging or ineffectively engaging in self-care monitoring. Another belief was that certain medicines delivered through a certain route (e.g., intra-muscularly) tended to have 'more power' as compared to other routes of administration. This specific belief has been captured, in a prior, important sociological account of rural Indian customs and public health beliefs [11]. Some patients also held the belief that modern medicines produced more 'side-effects'. Some patients were also distressed with restricting fluid intake to < 1000 ml/day, since they felt it was a departure from the normative habit, thus affecting self-care maintenance. Overall, the mechanisms through which these beliefs affect long-term planning, motivation and self-efficacy require further exploration.

Passivity, as we have defined it in this study is a novel behavioural trait; one prior report [20] has defined passivity in the context of treatment decision-making in heart failure. In our study, this trait was consistently seen among female patients or among males who were alcohol dependent. Such patients may benefit from behavioural interventions that improve motivation to take responsibility for their own health. Indeed, in our participant cohort, patients who explicitly and spontaneously expressed responsibility for their own health, displayed better overall self-care and adherence. This could have the collateral effect of reducing caregiver stress. Patients also expressed feeling stigmatised at having to take long term medications (particularly insulin for diabetes) with resultant anxiety, although this did not lead to treatment default. Patients with limited financial means and low health literacy who, despite formidable constraints, exhibited relatively good self-care in terms of treatment adherence, displayed traits of optimism, adhering to medications out of gratitude to the physician, resilience, and confidence. Such patients also had adequate social support from family or neighbours from whom financial or logistical assistance could be sought.

The 'intermediate factors' that we identified, namely, 'patient expectations' and 'hospital/ provider hopping' are novel findings in our study. We have chosen to call these intermediate

factors since they did not immediately shape self-care behaviour in our participant cohort. However, participants verbally expressed the fact that the thwarting of their expectations in terms of symptom recurrence or a re-hospitalization despite being adherent to treatment, left them feeling doubtful about the efficacy of treatment and that they anticipated switching to other treatments, including alternative medications in future. Hospital hopping may lead to a situation which culminates in the patient seeking specialized care at a tertiary care centre. However, since the patient did not have a trusted relationship with any hospital system or a provider, they did not have a clear action plan in the event of symptom exacerbations. This behaviour, representing a negation of trust, may be contrasted by the previously published finding in our facilitators of self-care report, where patients from lower socio-economic strata who had a trust-worthy relationship with their physician adhered to long term treatments despite financial constraints [12]. Vesting trust in the healthcare system and in the treating team is thus a pivotal attribute of patients and families who display optimum self-care, among Indian patients.

## Socio-cultural underpinnings of our findings

We also demonstrate that at the foundations of these outcomes are processes that are socially constructed, rooted in cultural and sociological factors that ultimately manifest in culturally variable behavioural traits and attitudes. Attribution of meaning, notions of significance, normative auras around certain behaviours, and internalisation are developed by individuals in coordination with others and are formulated within a social realm [21,22]. It has also been established that local-level understandings of illness, or health in general, are culturally modelled even if they are set within structural fact [23]. Physical illness–including its experience–and resultant medical care are intricately influenced by social aspects including gender, to the extent that even knowledge and assumptions about illnesses, and broad aspects of physiology as such, are socially constructed [24]. For instance, women inevitably experience significantly greater health strains compared to men due to these socially constructed roles around gender (and traditional gender patters in division of household labor), which interact and condition all components of caregiving, normalizing differential experiences between women and men in healthcare [25]. Further, it is known that the role of being 'sick' is more socially acceptable for women (germinating from gender roles and statuses, and resultant life circumstances for women), which then extends to even associated health-related behaviors and responses [26]. Feminine role expectations, which especially include (unpaid) caregiving, invariably lead to negative consequences for women's well-being [27]. Men's caregiving (and by that, token care-receiving) on the other hand is greatly contingent on the characteristics of men's families (and not their social status, employment, or even earning)–because of which women are treated (and are conditioned to treat themselves) as kin-keepers for all their lives, whereas men's roles as kin-keepers depends on the agency of women in the family and on the equations of marriage [28].

Proceeding from this point, we encourage the reader to visualise a *frame*. This frame is a crucible within which perceptions, attitudes and notions are prepared, and a consensus is generated in that society around these notions. It is by investigating into the nature of this 'frame' in specific contexts, that we fully understand the importance of upbringing, expectations around relationships, definitions of 'care,' 'self,' 'fate,' 'karma,' and so on, within that context. Perceptions of the self, of the companion and their role in caregiving (be it a spouse or any other family member), and others (doctor, nurse, paramedic), are all born of the frame.

If we outline what is relevant to our study, we see an interesting overlap of three social constructs–(a) of the ailing self, (b) of the caregiver, and (c) of the illness itself. It is from this triple overlap that the revealed behavior, specifically the negative determinants listed in this study-

chiefly passivity- originates, and which is why we call these outcomes as socially constructed. There are a number of examples from the results that have socially constructed roots from within this frame:

(a) *Not wanting to bother the caregiver*: perceiving the caregiver as an individual with more important priorities over and above care of the self, which is modelled over gender expectations, more than others.

(b) *Perception of the ailing self as a burden*: perceiving oneself as an entity that has value only when productive and contributing to the family's income or to domestic responsibilities, devalued therefore when sickness interrupts productivity.

(c) *Role of spouse and children as caregivers or observers*: borne out of the sense of one's loss of agency and productivity during illness, which is also based greatly on patriarchal norms and gender expectations.

(d) *Notions around fatalism, as associated with illness*: that it is out of one's deeds in the past, possibly even of a prior life.

(e) *Normalization of cardiovascular ailment as part of ageing*: a product of the social construction of the idea of ageing, associated but contradictory to the burdensomeness of the ailing self.

(f) *Beliefs around systems of medicine*: modern medicine as commercially exploitative in nature, which are shaped by socially constructed notions around 'Western' materialistic and monetized inclinations towards life and work. A detailed deconstruction of these structures and processes is warranted and may be the subject of a future study based on the same results.

## Utility of the findings of this study

A recent paper [29] presents the case for incorporating cultural context into treatment plans for managing cardiovascular diseases. The findings of this study have been used to inform a theory-based intervention package to improve heart failure self-care, which will be evaluated through a randomised, controlled trial. Key components of the intervention plan are based on our findings from this study and include–(i) a brief questionnaire to elicit certain entrenched beliefs, knowledge, motivation, as well as passive behaviour; (ii) task shifting–this questionnaire will be delivered by a trained nurse or a lay health worker who will be trained to identify negative and positive determinants and document them; the health worker will employ strategies involving patients and caregivers to mitigate negative determinants, while reinforcing positive determinants to improve self-care and enhance trust in the healthcare system; systematic education on selfcare monitoring and maintenance will also be delivered (iii) a simple screening tool for mood, anxiety disorders and cognitive dysfunction delivered by the health worker and a referral to psychiatry where necessary; (iv) brief psychological interventions to improve motivation, goal setting and self-efficacy, monitoring and maintenance behaviour; (v) the health worker will be tasked with facilitating access to medication subsidy schemes for poor patients, prepare a plan for managing symptom deterioration and if the patient consents, assist with some long-term healthcare related financial planning.

## Strengths and limitations

Careful purposive sampling at different stages of the study ensured that we had diverse representation across gender, health literacy, geography and socio-economic levels and thus helped

capture factors related to self-care decision making across a broadly representative section of the Southern Indian population. A carefully structured questionnaire helped us elicit information rich interviews. The findings are limited by the fact that most patients who were interviewed were from Southern India, where culture and social stratification is different compared to Northern India. We excluded patients with neurological deficits who did not have caregivers around and therefore, their challenges with self-care remain unknown. Possibly, differing findings may have been elicited if the interviews were carried out in the community. Future research needs to explore the mechanisms by which these beliefs impact motivation and habits and in turn, self-care behaviour.

## Conclusion

This study represents the first such grounded theory exploration of factors affecting self-care in Indian heart failure patients. We demonstrated that self-care in CHF is a complex process determined by a confluence of negative, intermediate and positive determinants, the interplay of which determines self-care behaviour among Indian heart failure patients. These factors have their roots in socio-cultural processes including gender. Efforts to improve self-care in practice must factor in these determinants to optimise clinical outcomes in heart failure. Assessing for and addressing these determinants during clinical interactions through multi-factorial approaches may help improve self-care among Indian CHF patients, thus improving medication adherence and clinical outcomes.

## Acknowledgments

The authors gratefully acknowledge the contributions of Ms. Immaculate Mary Josephine, research nurse, who assisted with interview translations and with general study co-ordination and Dr.Dhiraj R.S. who assisted with Tamil translation of interviews. We would also like to acknowledge Ms. Sangeetha Poojary, Research Associate who assisted with transcribing the interviews.

## Author Contributions

**Conceptualization:** Deepak Y. Kamath, K. B. Bhuvana, Bradi B. Granger, Denis Xavier.

**Data curation:** Kiron Varghese.

**Formal analysis:** Deepak Y. Kamath, K. B. Bhuvana, Luke Joshua Salazar, Anant Kamath, Jyoti Idiculla, Shruthi Kulkarni, Bradi B. Granger.

**Funding acquisition:** Deepak Y. Kamath, Bradi B. Granger, Denis Xavier.

**Investigation:** Deepak Y. Kamath, K. B. Bhuvana, Kiron Varghese, Anant Kamath, Jyoti Idiculla, Prem Pais, Shruthi Kulkarni, Denis Xavier.

**Methodology:** Deepak Y. Kamath, K. B. Bhuvana, Luke Joshua Salazar, Anant Kamath, Prem Pais, Shruthi Kulkarni, Bradi B. Granger, Denis Xavier.

**Project administration:** Deepak Y. Kamath, Prem Pais.

**Resources:** Deepak Y. Kamath.

**Software:** Deepak Y. Kamath.

**Supervision:** K. B. Bhuvana, Kiron Varghese.

**Visualization:** Deepak Y. Kamath, K. B. Bhuvana.

**Writing – original draft:** Deepak Y. Kamath, K. B. Bhuvana.

**Writing – review & editing:** Deepak Y. Kamath, K. B. Bhuvana, Luke Joshua Salazar, Anant Kamath, Jyoti Idiculla, Prem Pais, Shruthi Kulkarni, Bradi B. Granger, Denis Xavier.

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
