## [Decision Letter · Decision Letter 0]

22 Sep 2020

PONE-D-20-14836

A qualitative, grounded theory exploration of the determinants of self-care behavior among South Asian patients with a lived experience of chronic heart failure.

PLOS ONE

Dear Dr. Kamath,

Thank you for submitting your manuscript to PLOS ONE. After careful consideration, we feel that it has merit but does not fully meet PLOS ONE’s publication criteria as it currently stands. Therefore, we invite you to submit a revised version of the manuscript that addresses the points raised during the review process.

I agree with the reviewers that your manuscript is of interest, but both raise important points that need to be addressed, which need to be adequately addressed.

We look forward to receiving your revised manuscript.

Kind regards,

Hans-Peter Brunner-La Rocca, M.D.

Academic Editor

PLOS ONE

Journal Requirements:

Reviewers' comments:

Reviewer's Responses to Questions

**Comments to the Author**

1. Is the manuscript technically sound, and do the data support the conclusions?

Reviewer #1: Partly

Reviewer #2: Yes

2. Has the statistical analysis been performed appropriately and rigorously? 

Reviewer #1: N/A

Reviewer #2: Yes

3. Have the authors made all data underlying the findings in their manuscript fully available?

Reviewer #1: Yes

Reviewer #2: Yes

4. Is the manuscript presented in an intelligible fashion and written in standard English?

Reviewer #1: Yes

Reviewer #2: Yes

5. Review Comments to the Author

Reviewer #1: Dear Author

I have read your paper with great interest, and I hereby provide you with an oversight of my main comments:

In General:

- Be consistent in used terminology:

For example

“South Asian” and Indian are used interchangeably. However, Pakistan, Nepal, Bhutan, Bangladesh,

Afghanistan, Sri Lanka and the Maladives are also part of South Asia. It can be argued the results of your

research are not applicible to heart failure patients in these countries since

1) these countries have different cultures and thus it is possible HF-patients in these countries meet

other barriers and facilitators to self-care behavior

2) It is very difficult to generalise results of qualitative research.

Therefore it is not clear to me if the results are transferable to other South Asian countries and I suggest you

use the "Indian" in stead of "South Asian"

- Please check with author guidelines if standard British English of American English has to be used.

- Please perform a thourogh spell check of the entire document.

Title

A Qualitative grounded theory exploration of the determinants of self-care behavior, among South Asian

patients with a lived experience of chronic heart failure

This title suggests all determinants of self-care behaviour are thourghly discussed within the paper, however

this is not interiorly the case. The main focus of the paper is the barriers of self-care. Facilitators are briefly

discussed and published elsewhere.

Furthermore, this title also suggests the research has taken place in the whole of South Asia, while this is not

the case. (see also feedback above)

Abstract

Last sentence (line 80) it is stated that addressing these determinants may help to improve medication

adherence. What about adherence to other self-care advice such as fluid retention or salt?

Introduction

Please provide a reference for the statement on pag. 3, line 99 and 100: "Adherence to treatments is key to

reducing the risk of….."

pg. 4, Line 130: "Factors, principally, gender, the geography and society of their up-bringing, would shape

patients belief"

I do not understand the use of “would” in this sentence. Do or don't these factors shape patients beliefs?

Methods

Within the methods you describe on the one hand a written consent was obtained from all participants. On the

other hand you state in your results (pag 19, line 475) rural poor illiterate or semi-literate patients also

participated in your research.

Therefore I am wondering how you informed these patients? Did you use the same consent form for all

patients or was your consent form tailored to the literacy levels of your patients? Or, if it wasn't tailored, how

did you inform and obtained consent of these patients. Please state this in your answer.

Results

In general I find the results are presented in a complicated fashion:

- Titles and subtitles are difficult to distinguish from one another

- Terminology used in table 2 is not always used in the description of these themes and codes below. An

example:

Theme 1; entrenched beliefs:

In table 2 the first code of this theme is symptom/disease abstractions. In the description below you use the

term causal abstractions.

Please check terminology in the whole document and adjust.

- Furthermore, in the description of themes and codes, new codes such as “trust deficit or

denial/normalization who aren’t shown in table 2, are presented. Please integrate them in table 2

- Pag. 9, line 256: please place the parenthesis between 14 and 63.6% instead of in front of 14

- Pag. 15, line 385: “passive receipt of information”. I assume the authors mean “information seeking

passively” since this term is used in table 2. Please note there is a difference between reception of information

and seeking information. Check which one of these terms it is you mean and be consistent.

- Pg 16, paragraph II) intermediate factors: you state they modulate the selfcare process. Please provide a

reference for this claim.

Discussion

- Pg 21, line 536 (and further) authors state that patients who explicitly and spontaneously expressed

responsibility for their own health, displayed better overall self-care and adherence

I would like to bring the authors atention to the fact that this is new information. If this is a result of previous

research please provide the appropriate reference supporting this claim.

- Pg 22, line 561: authors state they see an interesting overlap of three social constructs….. please provide a

reference supporting the information considering social constructs

With kind regards

Reviewer #2: The manuscript addresses an important topic: the self-care behavior of patients, especially focusing patients in South Asia. The study uses a grounded theory approach to examine determinants of self-care. The treatment adherence is considered a main part of the self-care process. The self-care concept includes aspects of monitoring, maintenance and management of the disease.

The methodical approach is well explained and comprehensible. 22 patients and 17 caregivers were interviewed. Results were classified into 3 main categories that include several sub-categories. The negative determinants and the intermediate factors are the main focus of this article. The negative effects of self-care include entrenched beliefs, passivity, lack of knowledge, negative emotions, financial difficulties, and fatalism.

Intermediate factors are patients’ expectations, and provider hospital hopping.

The findings contribute to the further improvement and support of patients’ self-care by describing the importance of socio-cultural determinants and analyzing those determinants that are specifically important for patients within South Asia.

I have one main comments on the paper: Within the discussion, the authors draw an important link and encourage the reader to understand the results from a sociological perspective. However, the link to theories and current studies of medical sociology is missing. E.g. the mentioned passivity, especially of woman (line 532) could related to the cultural role of woman and social structures in India, which were certainly examined in further studies that not necessarily concern patients with HF but also consider other chronic conditions or general behavior patterns. The aspects considered as “socially constructed roots” (line 567-583) are very interesting but should also be linked to existing study results.

The following comments should be understood as suggestions and minor issues.

• Unfortunately, possible differences between the perception of patients and caregivers are not reported. Maybe the issues addressed in the methodology section: coping with evolving or caring for their own health (line 183, 184) could be elaborated.

• The first paragraph of section “results” (line 245-250) could be rephrased for a better understanding, e.g. line 247 – “of these” could not only refer to the patients included in the study could but also refer to the patients that refused and caregiver that were unavailable. In the abstract n=39 (line 57) participants are mentioned. Whereas in line 247 n=30 and line 250, n=7 participants are mentioned.

• The gender of the caregivers could be reported too. Line 254 only reports female patients.

• Line 254 reports the mean age of the participants and states the number of female patients in the following half sentence. This change of focus could confuse the reader.

• Table 1:

o It would help to have the total n within the table.

o How was the data within the section “self-care practices” collected? Was it a questionnaire or was the information collected during the interviews? The interview guide only included edema, blood pressure and weight.

o In the end of the manuscript some gender-related aspects are reported. Maybe it would be of value to add the information already within the table, e.g. report the self-care practices according to the gender.

o Since the number of participants is small, the decimal places could be omitted.

• Table 2: Within the table 2 the punctuation marks should be checked; e.g. sometimes literal speeches are not finished.

• For me, the aspect “provide or hospital hopping” is rather an expression of the absence of confidence/trust in the provider and therefore the negation of the trust highlighted in the positive points (table 3). Maybe, the difference could be made clearer.

I hope my suggestions help to further improve the paper.

6. PLOS authors have the option to publish the peer review history of their article (what does this mean?). If published, this will include your full peer review and any attached files.

Reviewer #1: No

Reviewer #2: No

---

## [Author Response · Author response to Decision Letter 0]

2 Nov 2020

Note to the Academic Editor: 

Dear Editor,

We are very grateful to both reviewers who have made astute observations that have helped improve the quality of this manuscript. We have followed the journal’s guidelines for the requested revisions.

Reviewer #1: 

Comment #1) In General:- Be consistent in used terminology: For example - “South Asian” and Indian are used interchangeably. However, Pakistan, Nepal, Bhutan, Bangladesh, Afghanistan, Sri Lanka and the Maldives are also part of South Asia. It can be argued the results of your research are not applicable to heart failure patients in these countries since :

- these countries have different cultures and thus it is possible HF-patients in these countries meet

other barriers and facilitators to self-care behavior.

- it is very difficult to generalise results of qualitative research.

Therefore it is not clear to me if the results are transferable to other South Asian countries and I suggest you use the "Indian" instead of "South Asian".

Response #1 – We agree with this important point. Owing to the significant cultural (ethnic, religious, socio-economic) diversity of the South Asian population, the results of this study may not be generalizable to the whole of South Asia. Accordingly, we have changed the terminology from South Asia to India in all relevant places, i.e, the title of the manuscript and in those sections dealing with study methods, results, discussion and conclusion, both in the abstract and the main paper. 

Comment #2) Please check with author guidelines if standard British English or American English has to be used; please perform a thorough spell check of the entire document. 

Response #2 – Spell check and corrections have been made at relevant places. 

Comment #3) Title - A Qualitative grounded theory exploration of the determinants of self-care behavior, among South Asian patients with a lived experience of chronic heart failure. 

This title suggests all determinants of self-care behaviour are thoroughly discussed within the paper, however this is not entirely the case. The main focus of the paper is the barriers of self-care. Facilitators are briefly discussed and published elsewhere. 

Response #3 – Thank you for clarifying. While selfcare facilitators are published elsewhere, that paper is a ‘research note’, which meant that we could not discuss (in the Discussion section of that manuscript) the impact of facilitators on selfcare, neither could we examine the findings in it’s socio-cultural perspective as we have discussed in this manuscript. Furthermore, we have listed the facilitators in Table 3 and discussed the impact of both barriers and facilitators in this paper. Hence, we do believe that the title is appropriate (with South Asian patients changed to Indian), since all emerging factors affecting selfcare have been discussed in this paper. 

Comment #4) Furthermore, this title also suggests the research has taken place in the whole of South Asia, while this is not the case. (see also feedback above)

Response #4 – Changes have been made as advised. 

Comment #5) - Abstract - Last sentence (line 80) it is stated that addressing these determinants may help to improve medication adherence. What about adherence to other self-care advice such as fluid retention or salt? 

Response #5 – I have replaced the word ‘medication’ with ‘treatment’. 

Comment #6) – Introduction - Please provide a reference for the statement on pag. 3, line 99 and 100: "Adherence to treatments is key to reducing the risk of….."

Response #6 - I have added the relevant reference. 

Comment #7) - pg. 4, Line 130: "Factors, principally, gender, the geography and society of their up-bringing, would shape patients belief"; I do not understand the use of “would” in this sentence. Do or don't these factors shape patients beliefs? 

Response #7 – We have deleted ‘would’, since these factors do influence selfcare. 

Comment #8) – Methods - Within the methods you describe on the one hand a written consent was obtained from all participants. On the other hand you state in your results (page 19, line 475) rural poor illiterate or semi-literate patients also participated in your research. Therefore I am wondering how you informed these patients? Did you use the same consent form for all patients or was your consent form tailored to the literacy levels of your patients? Or, if it wasn't tailored, how did you inform and obtained consent of these patients. Please state this in your answer. 

Response #8 – Thank you for noting this. We would like to clarify that the same consent form, translated into the respective vernacular language (through a professional translating agency) was used for these patients; we did not have a scaled down version of the form. There was one female patient who could not read the form at all. In her case, we involved the patient’s caregiver who knew how to read the language and explained study details and procedures to the patient and the caregiver and cleared their doubts. The patient’s thumb impression was obtained on the consent form along with the caregiver’s signature, followed by the investigator’s signature. 

Other patients who were semi-literate (basic reading and signing ability) were asked to read the form to the best possible extent, if needed with caregiver’s help. The investigator spent a longer duration with these participants to explain study details and clear doubts. These patients then signed the form, followed by the investigator’s signature. 

We have added the following sentence to the manuscript, Page 4, Line 144 – “Informed consent forms were translated into vernacular languages by a professional translating agency and approved by the ethics committee. Written informed consent was obtained from all participants, including semi-literate and illiterate participants in accordance with the Indian Council of Medical Research’s (ICMR) National Ethical Guidelines for Biomedical and Health Research involving Human Participants 2017.”

Comment #9) – Results - In general I find the results are presented in a complicated fashion: - Titles and subtitles are difficult to distinguish from one another.

Response #9 – We agree with your comment. However, we have attempted to simplify the format of presentation of results as far as possible. We hope that results presented in Table 2 will give readers a clear snapshot of themes and their supporting codes. 

Comment #10) - Terminology used in table 2 is not always used in the description of these themes and codes below. An example: Theme 1; Entrenched beliefs: In table 2, the first code of this theme is symptom/disease abstractions. In the description below you use the term causal abstractions. Please check terminology in the whole document and adjust. 

Response #10 – We sincerely apologize for this oversight and thank you for pointing this out. I have made the required corrections. 

Comment #11) - Furthermore, in the description of themes and codes, new codes such as “trust deficit or denial/normalization who aren’t shown in table 2, are presented. Please integrate them in table 2. 

Response #11 – Thank you, we have integrated these codes and extracts. 

Comment #12) Pag. 9, line 256: please place the parenthesis between 14 and 63.6% instead of in front of 14. 

Response #12 – Thank you, I have fixed this. 

Comment #13) - Pag. 15, line 385: “passive receipt of information”. I assume the authors mean “information seeking passively” since this term is used in table 2. Please note there is a difference between reception of information and seeking information. Check which one of these terms it is you mean and be consistent. 

Response #13 – Thank you. Passive receipt of information in more appropriate; since patients exhibiting passivity do not seek information actively, instead they do not seek information at all or they ‘chance’ upon the information. I have made the changes accordingly. 

Comment #14) - Intermediate factors: you state they modulate the selfcare process. Please provide a reference for this claim.

Response #14 – This is an empirical finding emerging from the dataset and memo. We have added a section about how these factors affected selfcare in our patient population. 

Added, Page 22, Discussion section, Line 546 – “The ‘intermediate factors’ that we identified, namely, ‘patient expectations’ and ‘hospital/provider hopping’ are novel findings in our study. We have chosen to call these intermediate factors since factors did not immediately shape selfcare behaviour in our participant cohort. However, participants verbally expressed the fact that the thwarting of their expectations in terms of symptom recurrence or a re-hospitalization despite being adherent to treatment, left them feeling doubtful about the efficacy of treatment and they anticipated switching to other treatments, including alternative medications in future. Hospital hopping may lead to a situation which culminates in the patient seeking specialized care at a tertiary care centre. However, since the patient did not have a trusted relationship with any hospital system or a provider, they did not have a clear action plan in the event of symptom exacerbations.”

Comment #15) – Discussion - Pg 21, line 536 (and further) authors state that patients who explicitly and spontaneously expressed responsibility for their own health, displayed better overall self-care and adherence. I would like to bring the authors attention to the fact that this is new information. If this is a result of previous research please provide the appropriate reference supporting this claim. 

Response #15 – This information is new and unique to this study. We have made the following addition – Page 22, Line 536, Indeed, “in our participant cohort”, patients who….”.

Comment #16) Pg 22, line 561: authors state they see an interesting overlap of three social constructs….. please provide a reference supporting the information considering social constructs. 

Response #16 – These are findings that emerge from the data, hence we have not referenced it. 

Reviewer #2: 

The manuscript addresses an important topic: the self-care behavior of patients, especially focusing on patients in South Asia. The study uses a grounded theory approach to examine determinants of self-care. The treatment adherence is considered a main part of the self-care process. The self-care concept includes aspects of monitoring, maintenance and management of the disease.

The methodical approach is well explained and comprehensible. 22 patients and 17 caregivers were interviewed. Results were classified into 3 main categories that include several sub-categories. The negative determinants and the intermediate factors are the main focus of this article. The negative effects of self-care include entrenched beliefs, passivity, lack of knowledge, negative emotions, financial difficulties, and fatalism.

Intermediate factors are patients’ expectations, and provider hospital hopping.

The findings contribute to the further improvement and support of patients’ self-care by describing the importance of socio-cultural determinants and analyzing those determinants that are specifically important for patients within South Asia.

Comment #1 - I have one main comment on the paper: Within the discussion, the authors draw an important link and encourage the reader to understand the results from a sociological perspective. However, the link to theories and current studies of medical sociology is missing. E.g. the mentioned passivity, especially of woman (line 532) could related to the cultural role of woman and social structures in India, which were certainly examined in further studies that not necessarily concern patients with HF but also consider other chronic conditions or general behavior patterns. The aspects considered as “socially constructed roots” (line 567-583) are very interesting but should also be linked to existing study results.

Response #1 – Thank you for identifying this. We have added the following section along with key references, which we hope will help put the results in better perspective. 

Addition – (Discussion Section) Physical illness – including its experience – and resultant medical care are intricately influenced by social aspects including gender, to the extent that even knowledge and assumptions about illnesses, and broad aspects of physiology as such, are socially constructed(24). For instance, women inevitably experience significantly greater health strains compared to men due to these socially constructed roles around gender (and traditional gender patters in division of household labor), which interact and condition all components of caregiving, normalizing differential experiences between women and men in healthcare (25). Further, it is known that the role of being ‘sick’ is more socially acceptable for women (germinating from gender roles and statuses, and resultant life circumstances for women), which then extends to even associated health-related behaviors and responses (26). Feminine role expectations, which especially include (unpaid) caregiving invariably lead to negative consequences for women's well-being (27). Men's caregiving (and by that token care-receiving) on the other hand is greatly contingent on the characteristics of men's families (and not their social status, employment, or even earning) – because of which women are treated (and are conditioned to treat themselves) as kin-keepers for all their lives, whereas men's roles as kin-keepers depends on the agency of women in the family and on the equations of marriage (28).

The following comments should be understood as suggestions and minor issues: 

Comment #2 - Unfortunately, possible differences between the perception of patients and caregivers are not reported. Maybe the issues addressed in the methodology section: coping with evolving or caring for their own health (line 183, 184) could be elaborated. 

Response #2 – Thank you for this very pertinent comment. Caregiver interviews were conducted to mainly support themes emerging from patient data. However, we did want to add a section on caregiver’s coping with fluctuating conditions and how they care for their own health. We thought the better of it since the Results and Discussion sections were already long and complex. We intend to publish the results from caregiver interviews as a separate brief ‘research note’ at a later date. 

Comment #3 - The first paragraph of section “Results” (line 245-250) could be rephrased for a better understanding, e.g. line 247 – “of these” could not only refer to the patients included in the study could but also refer to the patients that refused and caregiver that were unavailable. In the abstract n=39 (line 57) participants are mentioned. Whereas in line 247 n=30 and line 250, n=7 participants are mentioned. 

Response #3 – Thank you for pointing this out. We made an inadvertent error in reporting screening numbers (27 instead of 25, with 3 refusing consent). I have further clarified how we arrived at n = 39, i.e, 22 patients and 17 caregivers were interviewed. Additions to the manuscript have been made at relevant places. 

Comment #4 - The gender of the caregivers could be reported too. Line 254 only reports female patients.

Response #4 – Thank you, we have made the following addition – 

Added: (Results section, para 2, line 262) - 7 (41.2%) caregivers were male caregivers, while 10 (58.8%) were female caregivers.

Comment #5 - Line 254 reports the mean age of the participants and states the number of female patients in the following half sentence. This change of focus could confuse the reader. 

Response #5 – We have added mean age “of patients” to clarify this. 

Comment #6 - Table 1: It would help to have the total n within the table.

Response #6 – Thanks for identifying this; I’ve added total N into the table. 

Comment #7 - How was the data within the section “self-care practices” collected? Was it a questionnaire or was the information collected during the interviews? The interview guide only included edema, blood pressure and weight.

Response #7 – We had a separate brief data collection form to collect demographic and clinical data. It included a questionnaire with items pertaining to basic selfcare practices and dichotomous responses (Yes/No). 

Comment #8 - In the end of the manuscript some gender-related aspects are reported. Maybe it would be of value to add the information already within the table, e.g. report the self-care practices according to the gender. 

Response #8 – We have described this to the best possible extent (given the manuscript length) in the Results section titled, ‘Memo on gender, geographic, societal factors and social discourses shaping self-care’.

Comment #9 - Since the number of participants is small, the decimal places could be omitted.

Response #9 – We have adjusted to one decimal point, since the sample size is small. 

Comment #10 - Table 2: Within the table 2 the punctuation marks should be checked; e.g. sometimes literal speeches are not finished. 

Response #10 – I have corrected this to the best of my ability. 

Comment #10 - For me, the aspect “provide or hospital hopping” is rather an expression of the absence of confidence/trust in the provider and therefore the negation of the trust highlighted in the positive points (table 3). Maybe, the difference could be made clearer. 

Response #10 – Thank you for this astute observation. I have added the following in Discussion, Line 565 – “This behaviour, representing a negation of trust, may be contrasted by the previously published finding in our facilitators of selfcare report, where patients from lower socio-economic strata who had a trustworthy relationship with their physician adhered to long term treatments despite financial constraints(12). Vesting trust in the healthcare system and in the treating team is thus a pivotal attribute of patients and families who display optimum selfcare, among Indian patients.”

Added to manuscript - Hospital hopping may lead to a situation which culminates in the patient seeking specialized care at a tertiary care centre. However, since the patient did not have a trusted relationship with any hospital system or a provider, they did not have a clear action plan in the event of symptom exacerbations.

I hope my suggestions help to further improve the paper.

---

## [Decision Letter · Decision Letter 1]

25 Nov 2020

PONE-D-20-14836R1

A qualitative, grounded theory exploration of the determinants of self-care behavior among Indian patients with a lived experience of chronic heart failure.

PLOS ONE

Dear Dr. Kamath,

Thank you for submitting your manuscript to PLOS ONE. After careful consideration, we feel that it has merit but does not fully meet PLOS ONE’s publication criteria as it currently stands. Therefore, we invite you to submit a revised version of the manuscript that addresses the points raised during the review process.

Both reviewers agree that the revised manuscript has significantly improved. There are, however, still some remaining issues. Please note that many of them are related to some (minor) uncertainties of the text and some (spelling) mistakes. I, therefore, would like to ask to not only correct these points, but also to very carefully read again your whole manuscript before re-submitting. Please note that PLOS ONE does not provide proofreading of accepted manuscript. Therefore, the submitted manuscript must be ready for publication.

We look forward to receiving your revised manuscript.

Kind regards,

Hans-Peter Brunner-La Rocca, M.D.

Academic Editor

PLOS ONE

Reviewers' comments:

Reviewer's Responses to Questions

**Comments to the Author**

1. If the authors have adequately addressed your comments raised in a previous round of review and you feel that this manuscript is now acceptable for publication, you may indicate that here to bypass the “Comments to the Author” section, enter your conflict of interest statement in the “Confidential to Editor” section, and submit your "Accept" recommendation.

Reviewer #1: All comments have been addressed

Reviewer #2: (No Response)

2. Is the manuscript technically sound, and do the data support the conclusions?

Reviewer #1: Yes

Reviewer #2: Yes

3. Has the statistical analysis been performed appropriately and rigorously? 

Reviewer #1: Yes

Reviewer #2: Yes

4. Have the authors made all data underlying the findings in their manuscript fully available?

Reviewer #1: Yes

Reviewer #2: Yes

5. Is the manuscript presented in an intelligible fashion and written in standard English?

Reviewer #1: Yes

Reviewer #2: Yes

6. Review Comments to the Author

Reviewer #1: Dear Author

Thank you for your revisions, I have read the manuscript and have seen you have addressed my concerns. However, still some minor comments remain

1. Abstract

- line 48 and 50: you still use “South Asian” in stead of Indian. Is there a specific reason or is this an oversight?

2. Methods

- You state in line 194 and 195 that the interviews were performed in 7 languages. I am wondering if the principle interviewers (KDY and BKB) familiar with all of these languages? If not, how did you manage this?

- Line 237: please check the spelling of "nvivo"

- Line 240 en 244, figure 1: be coherent in the way you identify participants e.g. "patient 002" (line 240) and "patient_002" (line 244)

3. Results

- Line 307, table 2: please use a coherent manner to report the references to quotes. Sometimes references are placed in front of the quote, sometimes at the end of the quote and sometimes there are no references.

- Line 329: change 'self perception" to 'self-perception"

- Line 392: you state "Rather patients were passive...." I believe there is a word missing between "rather" and "patients"

4. Discussion

- line 579: please remove the additional blank space between "constructed" and "For instance..."

- Line 583: please remove the additional blank space between"healthcare" and "further"

- Line 629; you describe key components of an intervention plan while using the future simple tense. I agree to use this tense when you state you will develop an intervention plan. However I feel the present simple tense is better suited to describe the content of this intervention plan and to use in your enumeration. for example "key components will include (i) a brief questionnaire developed to elict...." in stead of " a brief questionnalire will be developed to elict..."

- Line 635; I don't think a health worker is part of your intervention plan. However, a simple screening tool for mood,...." is.

- Line 637: see comment line 629. again future simple is used in (iv) brief psychological interventions will be delivered....". While present simple is more approprioate to use in a enumeration. Therefore (iv) brief pyschological interventions to improve...." are part of this intervention plan.

With kind regards

Reviewer #2: Dear authors, Thank you for your corrections and answers to the questions. You answered most of the questions and the manuscript improved very much. But, from my point of view, it needs further improvement.

1. Regarding comment and response Nr. 7: Thank you for clarifying. When you used information that were derived from a questionnaire, you should mention this additional questionnaire within the method section.

2. Table 2 – columns of the table split the theme sections. So you don’t know if trust deficit and fatalism are new themes.

3. The quote “Nobody told me …” line 389 is differently assigned within table 2. In table 2 the quote is under the headline “Passive receipt of information” whereas in line 389 it is assigned to the one-way compliance. Personally, I don’t see the receipt of information (even a passive one) when I don’t get any information as the quote “Nobody told me.” would suggest.

4. The spelling of NVivo differs, e.g. line 225, 237 & 61 – please check spelling and consistency

5. The Article would be easier to read if you didn’t integrate the subtitles, e.g. “entrenched beliefs or notions” (line 309) or “passivity” (line 371), etc., into sentences. A paragraph after the theme would make the affiliation of the Codes to the themes clearer.

6. The terms in the table and the occurrence in the text should be in the same order, e.g. denial or normalization (line 359) is following the trust deficit in the text. In the table it follows the obstacle self-perception.

7. Sometimes, e. g. line 318 and 328 are :: - is that on purpose or a typo?

8. Line 355 signs for literal speech are missing.

9. Please use same terms and order in table 2 as in table 3, e.g. knowledge deficit vs. Lack of knowledge.

10. Regarding the social-cultural underpinning, it might be worth to have a look at the “Theory of planned behavior” and integrate it into the discussion. This theory relates attitudes, norms and perception into one behavioral framework.

Best regards

7. PLOS authors have the option to publish the peer review history of their article (what does this mean?). If published, this will include your full peer review and any attached files.

Reviewer #1: No

Reviewer #2: No

---

## [Author Response · Author response to Decision Letter 1]

5 Jan 2021

Dear Editor,

I would like to convey my gratitude to you and the reviewers for a thorough review of the manuscript. We have addressed all the comments that reviewers have pointed out. At least 2 of the co-authors reviewed the manuscript again for errors. 

Some of the minor typos that have been pointed out may not appear as track changes, but I assure you that we have corrected them.

Thanks again and I look forward to the next steps.

Sincerely,

Deepak

Reviewer #1:

1. Abstract

- line 48 and 50: you still use “South Asian” instead of Indian. Is there a specific reason or is this an oversight? 

Response: Thanks for your note. This was deliberate since we did indeed want to make the point that there are data which show that treatment adherence in CVD is particularly poor among the South Asian population. 

2. Methods

- You state in line 194 and 195 that the interviews were performed in 7 languages. I am wondering if the principle interviewers (KDY and BKB) familiar with all of these languages? If not, how did you manage this? 

Response: KDY has reading, speaking and writing proficiency in 4 languages (English, Hindi, Konkani, Kannada) and can understand a fifth language (Malayalam), while BKB is proficient in 3 (English, Telugu and Kannada). Our research nurse (Ms. Immaculate Sheela) is proficient in Tamil and Malayalam. When patients speaking Tamil and Malayalam interviewed patients, Ms. Immaculate would ask questions in the presence of one of the two principle interviewers, who would ask for on the spot translations if needed and direct further probe questions. 

- Line 237: please check the spelling of "nvivo" – 

Response: Thank you for noting and sincere apologies for the error. I have corrected it.

- Line 240 en 244, figure 1: be coherent in the way you identify participants e.g. "patient 002" (line 240) and "patient_002" (line 244) – Sincere apologies for the oversight; I have corrected the same.

Response: Apologies, I have corrected this.

3. Results

- Line 307, table 2: please use a coherent manner to report the references to quotes. Sometimes references are placed in front of the quote, sometimes at the end of the quote and sometimes there are no references.

Response: Thank you, I have made the necessary corrections.

- Line 329: change 'self perception" to 'self-perception"

Response: I have corrected this.

- Line 392: you state "Rather patients were passive...." I believe there is a word missing between "rather" and "patients"

Response: I have added “…mostly female..”

4. Discussion

- line 579: please remove the additional blank space between "constructed" and "For instance..."

Response: I have done so.

- Line 583: please remove the additional blank space between "healthcare" and "further".

Response: I have done so.

- Line 629; you describe key components of an intervention plan while using the future simple tense. I agree to use this tense when you state you will develop an intervention plan. However I feel the present simple tense is better suited to describe the content of this intervention plan and to use in your enumeration. for example "key components will include (i) a brief questionnaire developed to elict...." in stead of " a brief questionnalire will be developed to elict..."

Response: Thanks for this; I have corrected the tense everywhere.

- Line 635; I don't think a health worker is part of your intervention plan. However, a simple screening tool for mood,...." is.

Response: The trained health worker is currently playing a central, important role in the intervention plan. The PHQ-9 tool is administered by the health worker and a brief psychological intervention will be delivered along with a referral to psychiatry if needed.

- Line 637: see comment line 629. again future simple is used in (iv) brief psychological interventions will be delivered....". While present simple is more approprioate to use in a enumeration. Therefore (iv) brief pyschological interventions to improve...." are part of this intervention plan.

Response: I have made the change.

Reviewer #2: Dear authors, Thank you for your corrections and answers to the questions. You answered most of the questions and the manuscript improved very much. But, from my point of view, it needs further improvement.

1. Regarding comment and response Nr. 7: Thank you for clarifying. When you used information that were derived from a questionnaire, you should mention this additional questionnaire within the method section. 

Response: Thanks very much for noting this. I have added this under the Methods section, Para titled ‘Eligibility Criteria and sampling’, last sentence, “In addition, we captured basic clinical and demographic data and selfcare practices on a data collection form.” 

2. Table 2 – columns of the table split the theme sections. So you don’t know if trust deficit and fatalism are new themes.

Response: Sorry for this oversight. I have corrected this. 

3. The quote “Nobody told me …” line 389 is differently assigned within table 2. In table 2 the quote is under the headline “Passive receipt of information” whereas in line 389 it is assigned to the one-way compliance. Personally, I don’t see the receipt of information (even a passive one) when I don’t get any information as the quote “Nobody told me.” would suggest.

Response: Thanks for your note. This referenced patient had a long history of heart failure but said that she had not known that she had a cardiac problem for a long time. When we asked her if she had inquired of the doctors about the reasons for her symptoms/problems or whether she had tried finding it out from other sources, she replied back saying, “No, nobody told me; even the doctor did not tell me”, but while adhering to her medications as prescribed. However, this had come at the cost of many years of poor monitoring practice. In this context, we coded the statement both ways – (i) that the patient was content passively receiving information from her physician (passive receipt of information), and (ii) that she was content passively following basic treatment regimen advice (one-way compliance). 

For want of space, we have mentioned this only once under ‘one-way compliance’ in the main text. 

4. The spelling of NVivo differs, e.g. line 225, 237 & 61 – please check spelling and consistency

Response: I have made the correction.

5. The Article would be easier to read if you didn’t integrate the subtitles, e.g. “entrenched beliefs or notions” (line 309) or “passivity” (line 371), etc., into sentences. A paragraph after the theme would make the affiliation of the Codes to the themes clearer.

Response: Thank you for noting this; I have made the correction. 

6. The terms in the table and the occurrence in the text should be in the same order, e.g. denial or normalization (line 359) is following the trust deficit in the text. In the table it follows the obstacle self-perception.

Response: Sincere apologies for the error; I have corrected the same at relevant places. (Tables 2 and 3)

7. Sometimes, e. g. line 318 and 328 are :: - is that on purpose or a typo?

 Response: It was a part of transcribing convention that we had followed. However, in the interests of not confusing the readers, we have removed it from the relevant places. 

8. Line 355 signs for literal speech are missing.

Response: Thank you – have added it. 

9. Please use same terms and order in table 2 as in table 3, e.g. knowledge deficit vs. Lack of knowledge.

Response: Thank you – have corrected it.

10. Regarding the social-cultural underpinning, it might be worth to have a look at the “Theory of planned behavior” and integrate it into the discussion. This theory relates attitudes, norms and perception into one behavioral framework.

Response: I am very grateful for your suggestion. After your suggestion, I did refer to the Theory of Planned Behaviour. There are certainly overlaps between our findings and what the theory proposes. However, we are proposing a theory from our findings which we have called the ‘Extant Cycle Theory’. I have written up the paper as a ‘Brief Report’/ ‘Research Note’ and we will submit the paper around mid-Jan. Briefly, we’re proposing that the extant cycle theory is a type of ‘situation specific’ theory of selfcare in chronic disease and it has already informed the structure and delivery of the intervention package for the 3 centre RCT for which recruitment has just started. However, we shall now also put our findings in the context of the ‘Theory of Planned Behavior’ and evaluate whether changes are needed.

---

## [Editor Report · Decision Letter 2]

6 Jan 2021

A qualitative, grounded theory exploration of the determinants of self-care behavior among Indian patients with a lived experience of chronic heart failure.

PONE-D-20-14836R2

Dear Dr. Kamath,

We’re pleased to inform you that your manuscript has been judged scientifically suitable for publication and will be formally accepted for publication once it meets all outstanding technical requirements.

Kind regards,

Hans-Peter Brunner-La Rocca, M.D.

Academic Editor

PLOS ONE
---

## [Editor Report · Acceptance letter]

15 Jan 2021

PONE-D-20-14836R2 

A qualitative, grounded theory exploration of the determinants of self-care behavior among Indian patients with a lived experience of chronic heart failure. 

Dear Dr. Kamath:

I'm pleased to inform you that your manuscript has been deemed suitable for publication in PLOS ONE. Congratulations! Your manuscript is now with our production department. 

Kind regards, 

on behalf of

Dr. Hans-Peter Brunner-La Rocca 

Academic Editor

PLOS ONE